# Test and Treat Model for Tuberculosis Preventive Treatment among Household Contacts of Pulmonary Tuberculosis Patients in Selected Districts of Maharashtra: A Mixed-Methods Study on Care Cascade, Timeliness, and Early Implementation Challenges

**DOI:** 10.3390/tropicalmed9010007

**Published:** 2023-12-23

**Authors:** Palak Mahajan, Kathirvel Soundappan, Neeta Singla, Kedar Mehta, Amenla Nuken, Pruthu Thekkur, Divya Nair, Sampan Rattan, Chaturanand Thakur, Kuldeep Singh Sachdeva, Bharati Kalottee

**Affiliations:** 1International Union Against Tuberculosis and Lung Disease, South-East Asia Office, New Delhi 110016, India; amenla.nuken@theunion.org (A.N.); sampan.rattan@theunion.org (S.R.); chaturanand.thakur@theunion.org (C.T.); kuldeep.sachdeva@theunion.org (K.S.S.); bharati.kalottee@theunion.org (B.K.); 2Department of Community Medicine & School of Public Health, Postgraduate Institute of Medical Education and Research, Chandigarh 160012, India; kathirvel.s@pgimer.edu.in; 3National Institute of TB & Respiratory Disease, New Delhi 110030, India; n.singla@nitrd.nic.in; 4Department of Community Medicine, Gujarat Medical Education & Search Society Medical College, Vadodara 390021, India; kedar_mehta20@yahoo.co.in; 5Centre for Operational Research, International Union Against Tuberculosis and Lung Disease, 2 Rue Jean Lantier, 75001 Paris, France; pruthu.tk@theunion.org (P.T.); divya.nair@theunion.org (D.N.)

**Keywords:** latent TB infection, isoniazid preventive treatment (IPT), IGRA testing, close contacts, structured operational research training initiative (SORT IT), challenges, operational research

## Abstract

Tuberculosis preventive treatment (TPT) is an important intervention in preventing infection and reducing TB incidence among household contacts (HHCs). A mixed-methods study was conducted to assess the “Test and Treat” model of TPT care cascade among HHCs aged ≥5 years of pulmonary tuberculosis (PTB) patients (bacteriologically/clinically confirmed) being provided TPT care under Project Axshya Plus implemented in Maharashtra (India). A quantitative phase cohort study based on record review and qualitative interviews to understand the challenges and solutions in the TPT care cascade were used. Of the total 4181 index patients, 14,172 HHCs were screened, of whom 36 (0.3%) HHCs were diagnosed with tuberculosis. Among 14,133 eligible HHCs, 10,777 (76.3%) underwent an IGRA test. Of them, 2468 (22.9%) tested positive for IGRA and were suggested for chest X-ray. Of the eligible 2353 HHCs, 2159 (91.7%) were started on TPT, of whom 1958 (90.6%) completed the treatment. The median time between treatment initiation of index PTB patient and (a) HHC screening was 31 days; (b) TPT initiation was 64 days. The challenges in and suggested solutions for improving the TPT care cascade linked to subthemes were tuberculosis infection testing, chest X-ray, human resources, awareness and engagement, accessibility to healthcare facilities, TPT drugs, follow-up, and assessment. A systematic monitoring and time-based evaluation of TPT cascade care delivery followed by prompt corrective actions/interventions could be a crucial strategy for its effective implementation and for the prevention of tuberculosis.

## 1. Introduction

India plans to eliminate tuberculosis (TB) by 2025, a decade ahead of the 2035 global deadline [1]. Early diagnosis of TB, screening of contacts and other vulnerable groups, and offering TB preventive treatment (TPT) have been identified as important steps toward the elimination of TB [2]. It is estimated that TPT has the potential to reduce overall annual TB incidence rates by 8.3%, while the efficacy of currently available TPT drugs ranges between 60 and 90% [3,4,5]. Hence, with a vision to eliminate TB, in July 2021 the National Tuberculosis Elimination Programme (NTEP) of India released guidelines on the Programmatic Management of Tuberculosis Preventive Treatment (PMTPT) in India. The guidelines indicated an expansion of TPT beyond HHCs aged <5 years, i.e., household contacts (HHC) aged ≥5 years, after ruling out TB. Notably, they recommended TPT to people with TB infection (TBI) after ruling out active TB [6]. However, there is no gold standard test for diagnosing TB infection (TBI) and the choice of test is based on the cost, availability, human resources, and infrastructure [7].

There are two TPT care cascade models proposed and tested worldwide: (i) “Test and Treat” model (test and provide TPT only to HHCs with TBI); (ii) “Treat only” model (provide TPT to all eligible HHCs without TBI testing). Though both models are important, understanding the TPT care cascade in the “Test and Treat Model” is crucial since NTEP is expected to scale up this model across the country soon. Project Axshya Plus, a Global Fund-supported TPT model implemented by the International Union against Tuberculosis and Lung Disease (The Union), New Delhi, is supporting the Government of India in testing both these models in selected 108 districts of the country. Globally, major losses are observed at several steps of the TPT care cascade, which calls for a programmatic and systematic approach to address failures at each stage [6]. Though data on the coverage of testing for TBI are limited, research evidence from around the world indicates that 21% of patients with TBI who initiate TPT complete the treatment [8]. Between July and December 2021, India reported TBI testing among 1.5% of its eligible and vulnerable populations, of whom 30.6% tested positive for TBI and only 12% of eligible HHCs were started on TPT [1]. Though evidence is available on the “Treat only” model implemented among HHCs < 5 years, there is a lack of evidence on the “Test and Treat” model in India to understand the gaps/loss in the care cascade and the reasons for loss at each step of the care cascade to develop strategies for effective TPT implementation. In this context, we aimed to assess the TPT care cascade (“Test and Treat” model) among HHCs (aged ≥5 years) of index drug-sensitive PTB patients both clinically and microbiologically confirmed under the programmatic setting of selected Project Axshya Plus districts in Maharashtra. The specific objectives were to assess the (a) number (proportion) of HHCs screened and diagnosed for TB; (b) the number (proportion) tested for TBI; (c) the number (proportion) who tested positive; (d) the number (proportion) who initiated TPT; (e) the number (proportion) who completed TPT; (f) the average duration taken in each step of the TPT care cascade; and (g) to understand the perspectives of stakeholders about challenges and suggested solutions to improve the TPT care cascade. 

## 2. Materials and Methods

### 2.1. Study Design

This was a sequential, explanatory mixed-methods study with a quantitative phase (cohort study based on record review of routinely collected programmatic data) followed by a qualitative phase (descriptive study). 

### 2.2. Study Setting

#### 2.2.1. General Setting

Project Axshya Plus (Axshya- “A” is free and “xshya” is TB i.e., TB Free) was launched to strengthen TB preventive care among HHCs of TB patients in India (2021–24) [9]. The project is being implemented by the International Union Against Tuberculosis and Lung Disease (The Union), New Delhi, a Principal Recipient (PR) of a Global Fund grant which focuses on four essential interventions, namely (a) TPT; (b) multi-sectoral engagement; (c) public financial management system; and (d) operational research. The project covers 108 districts in seven states—Assam, Chhattisgarh, Himachal Pradesh, Jharkhand, Madhya Pradesh, Maharashtra, and West Bengal. TPT implementation is based on two treatment models: “Test and Treat” and “Treat only”. 

The “Test and Treat” model is being implemented in ten project districts across Maharashtra and Himachal Pradesh. Once an index patient with PTB is notified through the NIKSHAY portal (a web-based national TB notification system), field investigators visit the HHCs of the pulmonary tuberculosis (PTB) patient to screen the HHCs for the four cardinal symptoms of TB (cough for > two weeks, fever, weight loss, and night sweats). HHCs aged <5 years who are asymptomatic or not diagnosed with TB are started on TPT by a medical officer (MO) after ruling out contraindications. HHCs aged ≥5 years are evaluated for TBI using an interferon-gamma release assay (IGRA) in Maharashtra and IGRA or a tuberculin skin test (TST) in Himachal Pradesh. HHCs tested positive for TBI are evaluated using a chest X-ray (CXR) wherever available and a clinical assessment by an MO to rule out active TB disease and contraindications (chronic alcoholism, hepatitis, hepatotoxicity, peripheral neuropathy) for TPT. All eligible HHCs with TBI are provided TPT (isoniazid daily for six months, 5 mg/kg/day for those aged ≥10 years and 10 mg/kg/day for those aged <10 years) free of charge based on age and weight bands as per the national PMTPT guidelines.

#### 2.2.2. Specific Setting

Maharashtra is the second most populous state in the country, situated in the central part of India. The overall TB notification rate in the state in 2022 was 183.1 per 100,000 population. Of the notified cases, 86.9% were new cases and 66.8% were drug-susceptible pulmonary tuberculosis. Successful treatment outcomes were reported among 85.2% of TB patients from Maharashtra notified in 2021 [1]. Project Axshya Plus is strengthening TB preventive care in 33 out of 80 NTEP districts in the state. It is implementing the “Test and Treat” model in eight districts, namely Aurangabad Municipal Corporation (MC), Aurangabad Rural, Akola MC, Akola Rural, Buldana, Sangli MC, Sangli Rural, and Satara. IGRA testing was performed as part of testing for TBI in these districts before starting TPT. This study was conducted in all eight districts of Maharashtra where the “Test and Treat” model was implemented by Project Axshya Plus. As part of this project, all the services related to screening, IGRA testing, and chest X-ray are provided for free to all HHCs. 

In Maharashtra, the Catholic Health Association of India is the implementing partner for Project Axshya Plus. The district-level staff includes a team lead, a management information system (MIS) assistant, and 3–4 field coordinators (1 per 1000 PTB index patients). At the state level, the state technical advisor, a staff member of The Union, monitors and leads the program. Field investigators maintain daily data using an Excel-based Google sheet, which is consolidated by the MIS assistant for real-time monitoring and evaluation by the district, state, and national program management units. 

### 2.3. Study Population, Sample Size, and Sampling

*Quantitative:* All HHCs of patients with drug-sensitive PTB (both microbiologically/clinically confirmed) from selected “Test and Treat” districts of Maharashtra were diagnosed and initiated on anti-TB treatment (ATT) under NTEP from October 2021 to March 2022 (initial cohort of project implementation). 

*Qualitative:* Healthcare providers (assistant program manager, team lead, MO, field coordinator, and community health officer) and HHCs were conveniently selected for key informant (KII) or in-depth (IDI) interviews. Separate interview guides were used to conduct KIIs and IDIs. Information saturation decided the total number of interviews.

### 2.4. Data Collection, Variables, and Source

*Quantitative:* Data were extracted from routinely collected programmatic data of Project Axshya Plus. Following the removal of identification details, selected data variables from the study sites were downloaded from the Google sheet dedicated to Project Axshya Plus: index patient: age, gender, HIV status, diabetic status, and microbiological confirmation; HHCs: screening for active TB, treatment initiation for active TB, IGRA testing and result, CXR and result, TPT initiation, and TPT outcome. The dates of each step of TPT cascade care were also extracted, namely screening, IGRA testing, TPT initiation, and completion.

*Qualitative:* After obtaining consent, the healthcare providers were telephonically interviewed by P.M. (female, trained in qualitative research) in Hindi and audio-recorded using a call recorder application. An interview guide was used to understand the challenges and suggested solutions for improving the TPT care cascade among HHCs. HHCs were similarly interviewed by G.S.S. in Hindi using a separate interview guide and audio-recorded after obtaining consent to understand the challenges and suggested solutions in TPT cascade care. A total of 12 interviews were conducted with eight healthcare providers and four HHCs (two each stratified for completed and not completed TPT). Each interview lasted for an average of 25 min. 

### 2.5. Statistical Analysis 

*Quantitative:* The de-identified and cleaned Project Axshya Plus data available in the Microsoft Excel worksheet were exported and analyzed using STATA software (version 16.1). The variables (gender, age, symptom screening, diagnosis for TB, IGRA testing, IGRA test result, TPT initiation, TPT completion time, period between various steps of the TPT care cascade, and duration of TPT) were summarized using means (standard deviation, SD) or medians (interquartile range, IQR) and frequency (proportion) depending on the type and normality of the data. 

*Qualitative:* Final transcripts were prepared by P.M. on the same day based on the audio recording or verbatim notes of the interview. K.S. and P.M. conducted the manual descriptive content analysis (deductive coding) to generate codes and themes. Any disagreements between researchers were resolved by discussion. The results of the qualitative data were presented as a non-hierarchical thematic diagram. The qualitative research findings are reported as per the COREQ (Consolidated Criteria for Reporting Qualitative Research).

### 2.6. Ethics Approval

Ethics approval was obtained from the Ethics Advisory Group of the International Union Against Tuberculosis and Lung Disease, Paris, France. Verbal informed consent was obtained from participants before conducting the qualitative interviews.

## 3. Results

### 3.1. Quantitative 

A total of 4186 index PTB patients were registered and started on anti-tuberculosis treatment during the study period in the project area. Of these, 2514 (60.1%) were male and 2167 (51.8%) had microbiological confirmation of TB (Table 1). The mean age (SD) of the index PTB patients was 43.9 (18.4) years. The total number of HHCs of these index patients was 15,290. Of these, 7699 (50.3%) were female and the mean age (SD) of the HHCs was 31.3 (19.9) years. Of the total HHCs, 1118 (7.3%) were children aged <5 years. 

Of the total 14,172 HHCs aged ≥5 years, 65 (0.4%) HHCs reported symptoms suggestive of TB. Of those found to be TB-suggestive, 36 (55.4%) HHCs were diagnosed with active TB (Figure 1). Of all HHCs eligible for IGRA testing, 10,777 (76.3%) were tested and a total of 2468 (22.9%) were found positive for TBI. Of the total 10,777 HHCs tested, 5540 (51%) and 5237 (49%) were contacts of microbiologically and clinically confirmed PTB patients, respectively. IGRA positivity was found to be 26.3% and 19.3% among contacts of bacteriologically and clinically confirmed PTB patients, respectively. 

Of the total 2159 HHCs aged ≥5 years initiated on TPT, 1958 (90.6%) completed the same. A total of 201 (9.4%) could not complete TPT for various reasons. Of these, nine HHCs developed TB during the course of TPT.

The median (IQR) duration between treatment initiation of index PTB patients and screening of HHCs was 31 (14, 93) days (Table 2). Similarly, the median (IQR) duration between HHC screening and IGRA testing, and between HHC screening and TPT initiation, was 16 (3, 53) and 31 (14, 68) days, respectively. The median (IQR) duration to TPT completion was 183 (180, 191) days. 

HHC—household contact; IGRA—interferon-gamma release assay; IQR—interquartile range; PTB—pulmonary tuberculosis; TB—tuberculosis; TPT—tuberculosis preventive treatment. *An HHC is a person who shared the same enclosed living space as the index TB patient for one or more nights or for frequent or extended daytime periods during three months before the start of current TB treatment.

### 3.2. Qualitative

A total of 38 and 28 codes were deduced from the transcripts for challenges and suggested solutions to improve the TPT care cascade. 

#### 3.2.1. Challenges and Suggested Solutions to Improve HHC Screening, Testing, and Assessment for TPT

Figure 2 depicts the challenges and suggested solutions for improving the screening, testing, and assessment to initiate TPT among HHCs in the study area. The codes are categorized under the subthemes of TB testing, chest X-ray, accessibility issues, awareness and engagement, and human resources-related challenges and solutions. 

TBI testing: Delays in initiating TPT due to multiple sequential diagnostic procedures (screening, IGRA/TST, CXR, and pre-treatment assessment) before the initiation of TPT, labs located far away, and poor accessibility of X-ray facilities pose a serious challenge for testing, assessment, and TPT initiation. 


*“had to send samples to Mumbai; do not have any in-district labs; takes at least 72 h to get the test results”; “We screened and sampled them at home. But the HHC need to come to the hospital for X-ray crossing all his barriers”.*


To reduce the challenges related to TBI testing and chest X-rays, the healthcare providers suggested strengthening local laboratories and facilities in addition to using point-of-care testing kits, hand-held X-rays, and mobile X-ray vans. 


*“We can finish all the workup at home if there is a point of care testing and a mobile X-ray van that might reduce the time taken for work up”*


Awareness and engagement: The healthcare providers highlighted a difficulty in engaging the patients due to lack of symptoms among HHCs and low awareness. There was also a fear of disease, medical assessment for TPT, and TPT initiation both among eligible HHCs and their family members.


*“HHCs told that why are you asking us to take medicine when we don’t have any symptoms. Though TBI testing report helped in convincing them”.*


The healthcare providers suggested disseminating information at the community level through various media, providing adequate counseling to HHCs, and especially engaging the heads of households. In addition, poor awareness about TPT implementation and its guidelines among routine health staff and private doctors delayed the process, since the HHCs consulted these people to obtain more details.


*“our family doctor told that it (TPT) is not necessary”; “the community health officer is not aware of the TPT assessment”*


For the above challenges, it was suggested that training or continued medical education programs be provided to routine health staff and to private doctors.

Human resources: The allotment of HHCs of 1000 PTB patients to one field coordinator was felt as high burden by the healthcare providers. They suggested a reduction of this number to 1000 HHCs in view of the multiple visits needed before and during TPT initiation and completion. Further, they also shared that the workload was further aggravated by the recording of the same data in multiple formats.


*“quite overloaded with the work… over and above we need to enter the same data in excel, google sheet and also in diary…”; “fixing appointment and remembering the pending assessment for more than 5000 HHCs is quite hectic…it needs to be reduced to 1000 HHCs for each of us”*


Accessibility issues: Most of the HHCs worked and were hesitant to take leave to visit a health facility, due to which the HHCs suggested that the services be extended till evening and on weekends. Further, both HHCs and healthcare providers mentioned visiting multiple times for assessment and transport-related issues as one of the difficulties. 


*“I cannot take leave. I can come on Saturday or Sunday if the hospital is open”;*



*“Sometime I have to visit the HHC three times even to motivate them to visit health facility for chest x-ray but some HHC are unable to take leave from work to visit health facility”*


#### 3.2.2. Challenges and Suggested Solutions for TPT Initiation, Follow-up, and TPT Completion among HHCs

Figure 2 depicts the challenges and suggested solutions for TPT initiation, follow-up, and TPT completion identified in this study. The codes are categorized under the subthemes of TPT drugs, awareness and skills of healthcare staff, and TPT follow-up and adherence. 

TPT drugs: The healthcare providers opined that adverse events due to TPT and long duration of TPT (6H—isoniazid therapy for 6 months) were a major reason for poor adherence and completion of TPT. Some of them mentioned the non-provision of drugs to them to distribute to HHCs and having to manage minor adverse events as one of the challenges of making HHCs complete TPT. 


*“Of course, if patient is suffering from diarrhea again and again, nausea, vomiting or loss of appetite, so adherence become little difficult.”; “Patient (HHC) has to take it for six months. And I also have to follow them for six months, instead shorter regimen (3HP: 3 months of combination of isoniazid and rifapentine) can be given to all… that is for only 3 months”; “I called the patient (HHC) to collect the TPT for third month. But the drug was stocked out due to which patient has to wait for some days to collect his medicine (TPT).”*


For the above issues, they suggested spending adequate time with the HHCs to counsel them about TPT, its adverse events, and their management. In addition to always maintaining adequate stock, they also suggested starting wise boxes for the HHCs.

Awareness and skills: The healthcare providers and HHCs mentioned that poor awareness of drug interactions among medical officers at public hospitals and among private doctors was one of the difficulties in continuing and completing TPT among HHCs, for which they suggested adequate training. 


*“our doctor told to stop this medicine (TPT) for my son mentioning this is the reason for the current status (HHC is addicted to alcohol)”*


Follow-up and adherence: Multiple visits to collect TPT drugs, multiple health staff visits (and related confidentiality issues), non-issuing of drugs while moving to another address, and lack of testing at the end of TPT were challenges cited by the HHCs in continuing and completing TPT. It was suggested that incentives be provided to them, community health workers (CHWs) be involved, and drugs be issued while they are traveling.


*“I don’t have money to visit every month to collect the medicine.”; “HHCs concerned about the multiple visits to his home by various staff… they also asked me enquire about them only through direct phone calls to them or through Accredited Social Health Activist (ASHA). Not to enquire through villagers or neighbors.”*


The healthcare providers faced difficulties in assessing adherence (counting pills in blister packs) and faced misinformation by the HHCs on their intake of drugs and symptoms. 


*“our MIS call them every month to check whether they are taking drugs or not. Some patients (HHCs) don’t take it daily… but they (HHCs) lie to us”*


For the above challenges, they suggested optimizing follow-ups in addition to involving CHWs.

## 4. Discussion

The current study assessed the “Test and Treat” model of the TPT care cascade among HHCs (aged ≥5 years) of PTB patients from Project Axshya Plus in Maharashtra, India, using a mixed-methods design. The presence of symptoms suggestive of TB was reported in less than one percent of HHCs screened. More than three-fourths of the HHCs were tested for IGRA and nearly one in four tested positive for the same. All eligible HHCs with TBI were started on TPT, of whom more than 90% completed it. It took at least one month and two months (median duration), respectively, from ATT initiation among index pulmonary TB patients to screen the HHCs and to initiate TPT. Similarly, it took at least one month to initiate TPT from the time of HHC screening. The qualitative survey found that the challenges and suggested solutions related to TBI testing and chest X-rays were existing human resources, awareness and engagement, TPT drugs, follow-up and assessment, and accessibility of the services linking each step of the care cascade. 

To the best of our knowledge, this is the first study from India that assesses the “Test and Treat” model of the TPT care cascade among HHCs under programmatic settings. The current study screened all HHCs (aged ≥5 years) of index PTB patients. We found a suboptimal prevalence (<0.5%) of HHCs with symptoms suggestive of TB at the first screening step compared to the nationally reported prevalence (2%) under programmatic settings in 2022 [1]. More than half of symptomatic HHCs were diagnosed with TB in the current study, compared to the national average, which is one-fourth [10]. However, the same figure reported under NTEP for Maharashtra was nearly 50% in 2022 [10].

The low proportion of symptomatic HHCs could have been due to the fear and stigma associated with TB and mistrust of the healthcare system/healthcare providers, due to which the HHCs might have underreported their symptoms. 

In the “Test and Treat” model of the TPT care cascade, a little less than one-fourth of the HHCs missed IGRA testing in this study. Yet, the proportion of HHCs tested (>75%) with IGRA was higher than the globally reported pooled proportion (62.3%) [8]. However, the reasons for not getting tested are similar to observations reported across countries, like absence of symptoms, fear of procedures, mistrust of health facilities or healthcare providers, and difficulty in reaching the HHCs. Notably, it took at least a fortnight to sample the HHCs for IGRA testing, which may have been due to the busy schedules of the HHCs, the limited availability of community health staff, and the time taken to convince the HHCs. 

The prevalence of TBI in our study was 22.9%, which is a little less than the pooled prevalence of positive tests for TBI found across studies conducted in low- and middle-income countries (LMICs), which ranges from 41 to 69% [8,11,12,13,14]. This difference may be attributed to specific factors, including the inclusion of only HHCs of PTB patients (both microbiologically and clinically confirmed) and not of high-risk groups, and use of only IGRA, not TST. TST positivity is generally higher due to BCG vaccination in most LMICs. 

TBI infection was found to be higher among contacts of microbiologically confirmed PTB patients, who have been established to carry a higher risk of transmission [15]. Previous studies found microbiological confirmation of index patients to be associated with IGRA positivity [13,16]. Therefore, it is crucial to prioritize attention to and focus on HHCs of microbiologically confirmed PTB patients to diagnose and prevent the spread of TBI early.

Of the HHCs with TBI, one-eighth dropped out before undergoing a chest X-ray and medical assessment. This could have been due to X-ray accessibility issues (time, distance, and transport), the need for multiple or repeated visits, loss of work hours, perceived lower risk of disease, and poor awareness of the disease and TPT among HHCs and their family members [11,13,17]. Despite these dropouts, all eligible HHCs were started on TPT after the medical assessment. 

Though more than 90% among those who initiated TPT completed it, 0.4% were report reported to develop active TB disease. HHCs who dropped out at each step of the cascade, especially HHCs with abnormal chest X-rays and those who completed TPT, should have been evaluated further for TB and/or TBI, since there could be a high number of active TB cases hidden in this population, which needs to be studied. This is especially important since for more than half of the HHCs, TPT was initiated a month after the initial screening. This delay may have defeated the ultimate purpose of providing TPT to HHCs, especially in the immediate first two months from the diagnosis of the index PTB patient in the house. 

Our study had the following strengths. First, this study was conducted under programmatic conditions, reflecting a realistic scenario of the TPT care cascade. Second, it included all HHCs of index PTB patients initiated on TPT during the study period from all districts of Maharashtra where the “Test and Treat” model was implemented. Due to the above strengths, the findings of this study are potentially generalizable and scalable to other similar settings. Third, it used a mixed-methods study design that provided insights into the challenges and suggested solutions to improve performance at each step of the TPT care cascade. 

The current analysis is not without limitations, which must be considered while interpreting and generalizing the findings. First, the data used in this study were part of the first out of the six phases of the program’s implementation. Any implementation will obviously face some setbacks in the early days, but our use of these data will greatly help in setting strong grounds for the program through learning and experience. Second, at every stage of the TPT care cascade, there were data gaps, and no attempt was made to make up for them. However, the missing data were not systematic enough to affect the findings of this study. Similarly, this study could not trace the HHCs who were not tested with IGRA and the incidence of TB among them. Similarly, we could not assess the status of people with TBI who did not undergo a chest X-ray but completed TPT. Third, no adjusted analysis was carried out to assess the factors associated with completing each step of the TPT care cascade, since variables beyond gender and age were not routinely collected and not available. Fourth, the specific (quantitative) reasons for dropouts at each stage of TPT care cascade could not be gathered because of the limited programmatic data captured under Project Axshya Plus. Fifth, despite the large sample size, only four household contacts could be interviewed. There exists a scope for sufficiently gathering perspectives from various subgroups of HHCs so as to perform a within-group comparison, although healthcare providers did inform us on some of the challenges and solutions on their behalf. Since neither a comparison with the “Treat only” model nor a cost-effectiveness analysis were carried out, we cannot comment on whether the “Test and Treat” model is better or more cost-effective than the other model. This can be tested in future studies.

## 5. Conclusions

The current study assessed the “Test and Treat” model of the TPT care cascade among the HHCs of PTB patients from Project Axshya Plus. The reasons for the non-uptake of IGRA testing need further research, since nearly one-fourth of the HHCs dropped out at this step. Further, a follow-up system needs to be established among people who drop out at each step of the care cascade. Though all eligible HHCs were started on TPT, there is a need for systematic follow-up and real-time monitoring/supervision of cascade care delivery followed by prompt corrective actions/interventions to achieve improved outcome at each step of the care cascade. 

## Figures and Tables

**Figure 1 tropicalmed-09-00007-f001:**
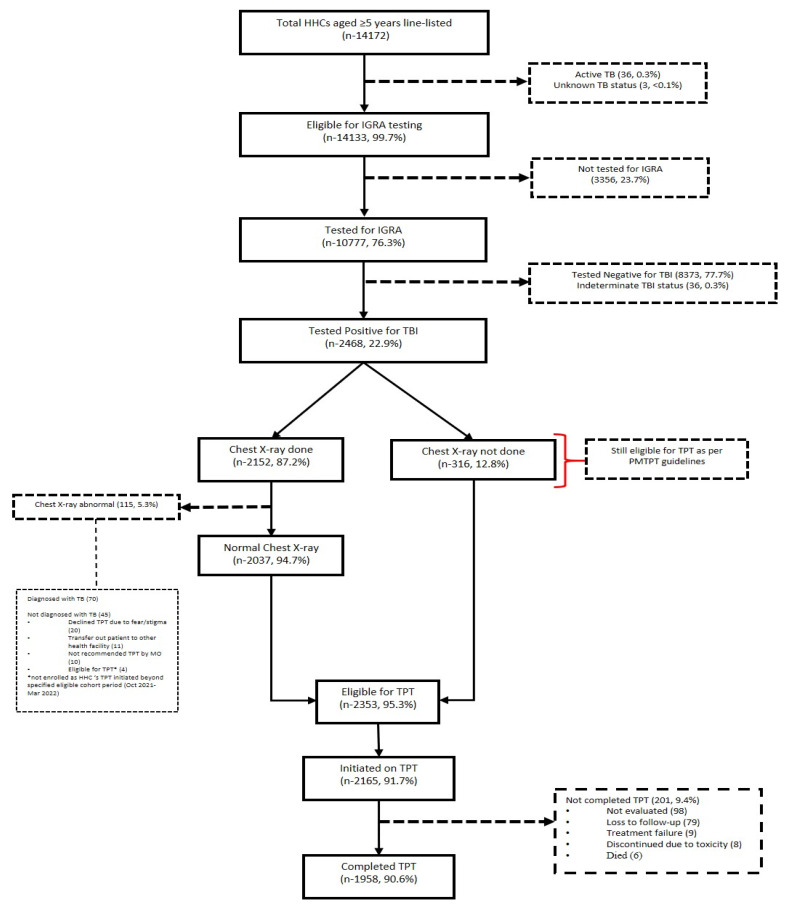
TPT care cascade of HHCs of PTB individuals (≥5 years) from “Test and Treat” model districts of Project Axshya Plus. Abbreviations: CXR—chest X-ray; HHCs—household contacts; IGRA—interferon-gamma release assay; TB—tuberculosis; TBI—tuberculosis infection; TPT—tuberculosis preventive treatment.

**Figure 2 tropicalmed-09-00007-f002:**
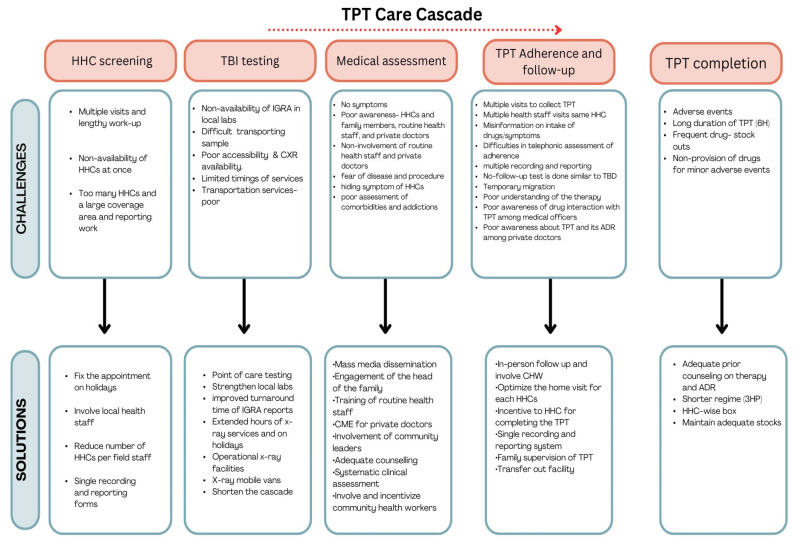
Challenges and suggested solutions to improve the TPT care cascade in “Test and Treat” model under Project Axshya Plus. Abbreviation: 6H—isoniazid therapy for 6 months; CHW—community health workers; CME—continuing medical education; HHC—household contact; IGRA—interferon-gamma release assay; TBD—tuberculosis disease; TPT—tuberculosis preventive treatment.

**Table 1 tropicalmed-09-00007-t001:** Demographic profile of index PTB patients and HHCs of index PTB patients from “Test and Treat” model districts of Project Axshya Plus.

	Index PTB Individuals		HHCs * of PTB Individuals	
Characteristics	n	%	n	%
Total	4186	100	15,290	100
Age (years)				
0–4	-	-	1118	7.3
0–14	83	1.9	2625	17.2
15–24	731	17.5	2749	17.9
25–34	592	14.2	4536	29.7
35–44	801	19.2	101	0.6
45–54	649	15.5	1770	11.6
55–64	540	12.9	1233	8.1
≥65	790	18.9	1158	7.6
Not recorded	4	0.1	-	-
Gender				
Male	2514	60.1	7591	49.7
Female	1671	39.9	7699	50.3
Transgender	1	0.02		
Microbiological confirmation				
Yes	2167	51.8	-	-
No	2015	48.1	-	-
Not recorded	4	0.1	-	-
Number of HHCs				
1–3	-	-	4807	31.4
4–6	-	-	7020	45.9
6–9	-	-	2299	15.0
>9	-	-	1164	7.6

All are column percentages. Abbreviations: HHC—household contact; PTB—pulmonary tuberculosis. * An HHC is a person who shared the same enclosed living space as the index TB patient for one or more nights or frequent or extended daytime periods during three months before the start of current TB treatment.

**Table 2 tropicalmed-09-00007-t002:** Duration between various steps of TPT care cascade among HHCs of PTB patients from “Test and Treat” model districts of Project Axshya Plus.

Duration (in Days) between	Median	IQR
Treatment initiation of index patient and HHC screening (n-12,195/14,172)	31	(14, 93)
HHC screening and IGRA testing (n-10,777/10,777)	16	(3, 53)
IGRA testing and TPT initiation (n-2199/2468)	12	(7, 23)
Treatment initiation of index patient and TPT initiation (n-1648/2468)	64	(35, 107.8)
HHC screening and TPT initiation (n-2150/2165)	31	(14, 68)
Initiation and completion of TPT (n-1926/1958))	183	(180, 191)
HHC screening and initiation of TB treatment (n-26/40)	25	(15.5, 69.8)

## Data Availability

Requests to access these data should be sent to the corresponding author.

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
