# Peer review of "Test and Treat Model for Tuberculosis Preventive Treatment among Household Contacts of Pulmonary Tuberculosis Patients in Selected Districts of Maharashtra: A Mixed-Methods Study on Care Cascade, Timeliness, and Early Implementation Challenges"

_tropicalmed, 2023, doi:10.3390/tropicalmed9010007_

Round 1

Reviewer 1 Report

Comments and Suggestions for Authors

Dear Authors,

Congratulations for a very important work on "Test and Treat LTBI". 

Few suggestions for further improvement of this manuscript

Minor corrections:

1. Abstract:  Line 28- number of TB patients can be added before the number of HHCs in the sentence.

2. Abstract: line 32 to 34- The sentence is not complete

3. Abstract- mention IGRA as method of testing for TBI as that would create more interest in readers as both IGRA and TST are used for TB infection testing, but with variable results. 

3. Settings: Line 113- 183.1 per lac population- change lac to 100,000 as the term lac is not used globally

Major issues:

1. Statistical Analysis: Line 158 to 163- the summary stats could be improved with at least sound univariate analysis ( multivariate could still be much better with this very important data you have)

2. Table 1- Age group 35-44 HHC number given as 101,  14.2%. This cant be right. 14% of 15290 is not 101. please correct

3. This study included HHCs of all Pulmonary TB patients, while most other studies use HHCs of only bacteriologically positive pulmanary TB. In this study almost half are clinically diagnosed. This is a good opportunity to analyse the differences in this two important groups.

What is the % of people who got tested for TBI among bacteriologically positive and clinical tb index cases?  

In this group how the TB infection prevalence varies?

Among the two groups is there any difference in % of people who accept TPT and also the treatment completion etc..

This is very important as the results could help even policy decisions, whether to test all pulmonary TB contacts or only bacteriologically positive  pulmonary TB. 

4. Fig-1- What happened to the 115 Chest X-Ray abnormal TBI positive people? How many diagnosed with TB? If not TB, whether included for TPT?

5. 90% TPT completion is very good compared to many other studies and programmes, even though the TPT was with 6 months of INH. What are the contributing factors? Compared to cohorts with no testing, but only treat policy, is the treatment completion better here? It is very likely that people would take drugs if they know that they have an infection with a positive test results than just asking people to take drugs, but if you have some data or references (there are many) on that, it would be an important message for test and treat. 

6. Discussion line 333- suboptimal prevalence of symptoms among HHC. Is this similar among contacts of bact positive and negative TB? The comments on low prevalence of symptoms due to fear /stigma etc- do you have data from the qualitative part of the study to support or any other relevant references?

7. The low prevalence (22.3%) of LTBI in HHC is not well explained. If you look at the LTBI prevalence among the two different groups mentioned earlier (contacts of bact positive and negative), that might give you some information for the explanation- if there is significant differences. 

8. Conclusion:  encourage you to give the key messages, like Test and Treat policy for LTBI works well, have good treatment completion, however there are delays etc... 

9. Finally- there are some good recent publications, many are from India, that you can reference - some links given below

https://onlinelibrary.wiley.com/doi/10.1111/tmi.13693#:~:text=associated%20with%20LTBI.-,Results,CI%3A%2050.1%E2%80%9355.1%25).

https://pubmed.ncbi.nlm.nih.gov/35927930/

https://pubmed.ncbi.nlm.nih.gov/33046016/

https://pubmed.ncbi.nlm.nih.gov/36740307/

Reviewer 2 Report

Comments and Suggestions for Authors

This is an interesting and well conducted study, involving two parts, one clinical/epidemiologic and one qualitative. 

The main question addressed by the research is to assess the Test and Treat strategy among household contacts of diagnosed tuberculosis cases in a large area in India within an organized national program.

The topic is relevant and adds significant information,indicating important areas for improvement of practices, not only to optimize the Test and Treat process for diagnosis and management of tuberculosis in household contacts, but also to assess the secondary rate of tb infections among household contacts and improve adherence of persons involved.

Analysis is basic as it covers the information needed to assess the effectiveneness of the study, but precludes delineation of the factors that might be associated them.

Although risk factor analysis is not performed, as these parameters were not recorded as part of the original strategy, the epidemiologic information provided is meaningful and provides areas for comparisons with other similar studies and strategies. This is a limitation that the authors mention and necessitates future studies to go in depth to this respect.

Conclusions and Discussion meet the study findings and provide implications for future studies. The arguments are well described and address the core question of the study and the test and treat strategies.

Reference list is updated. Authors could add some more references in the discussion section. While the narrative part of the manuscript is well written and complete, there seems to be a lack of references that can further support the discussions and arguments. I'd suggest the authors revise their reference list with more studies, particularly with other strategy types.

Tables and figures are appropriate and descriptive.

Reviewer 3 Report

Comments and Suggestions for Authors

This is an important study that can guide programs on scaling up TPT in high-TB prevalent settings with limited resources. Also, documents gaps in the cascade of testing and enrolling household contacts for TPT and, as such identifies important lessons learned for improvement and optimizing the strategy of test and treat. 

A few comments for consideration by line:

25- Add "regardless of bacteriological confirmation' as bacteriologically confirmed and clinically diagnosed PTB were included.

28: 39 were not eligible for TBI testing, suggest adding that among 14133 eligible for TBI testing 76.3% were tested.

34: "Accessibility" refers to geographic access and availability of CXR and IGRA in the facilities. I would suggest being specific on these issues.

35: Timing could be a more specific recommendation as TPT initiation is around two months and could be simplified if no test for TBI is performed or done at the same time or if CXR is done first as some of HHC might be asymptomatic. 

62: How many districts are in India? it seems there are 717 districts in total.

94: Were all districts of seven states included? if not how they have been selected?

114: add "drug-susceptible TB"

117: How these eight districts were selected (criteria) for the Test and treat model?

119-120: What was the eligibility criterion? symptomatic or regardless of symptoms?

128: How data was validated? were paper-based tools used for cross-checking?

132: Suggest to add "or clinically diagnosed"

Table 1: The proportion is high? more explanation on how clinically pulmonary TB is diagnosed is needed in the setting to understand this figure. 

222 -224: The issue of multiple sequential tests, geographical access, and availability of CXR and IGRA could be developed in discussion as implications of this research for improvement.

393: Limitation: Qualitative: the perspective of HHC was not considered and contrasted with the perspective of the health staff interviewed. 

Discussion: Implications of findings: optimize access to screening CXR more sensitive, using CAD, and both at the same time with IGRA and TST; including community members for qualitative.
